# Oral Arginine Supplementation in Healthy Individuals Performing Regular Resistance Training

**DOI:** 10.3390/healthcare11020182

**Published:** 2023-01-06

**Authors:** Ștefan-Sebastian Busnatu, Octavian Andronic, Maria-Alexandra Pană, Anca Pantea Stoian, Alexandru Scafa-Udriște, Nicolae Păun, Silviu Stanciu

**Affiliations:** 1Department of Cardiology, Carol Davila University of Medicine and Pharmacy, Bagdasar-Arseni Emergency Hospital, 050474 Bucharest, Romania; 2Department of Cardiology, Carol Davila University of Medicine and Pharmacy, University Emergency Hospital of Bucharest, 050098 Bucharest, Romania; 3Department of Diabetes, Nutrition and Metabolic Diseases, Carol Davila University of Medicine and Pharmacy, 020021 Bucharest, Romania; 4Department of Cardio-Thoracic, Carol Davila University of Medicine and Pharmacy, Emergency Clinical Hospital, 014461 Bucharest, Romania; 5Department of Cardiology, Carol Davila University of Medicine and Pharmacy, Theodor Burghele Clinical Hospital, 020021 Bucharest, Romania; 6Department of Cardiology, Carol Davila University of Medicine and Pharmacy, Central Military Hospital, 010825 Bucharest, Romania

**Keywords:** arginine, arterial stiffness, resistance exercise, cardiovascular risk factors

## Abstract

Resistance exercise training is well documented as having cardiovascular benefits, but paradoxically, it seems to increase arterial stiffness, favoring the development of high blood pressure. The present study investigates the potential effects of oral supplementation with arginine in healthy individuals performing exercise resistance training. We studied 70 non-smoking male subjects between the ages of 30 and 45 with normal or mildly increased blood pressure on ambulatory monitoring (for 24 h) and normal blood samples and echocardiography, who performed regular resistance exercise training for at least five years with a minimum of three workouts per week. They were divided into two groups in a random manner: 35 males were placed in the arginine group (AG) that followed a 6-month supplementation of their regular diets with 5 g of oral arginine powder taken before their exercise workout, and the control (non-arginine) group (NAG) consisted of 35 males. All subjects underwent body composition analysis, 24 h blood pressure monitoring and pulse wave analysis at enrollment and at six months. After six months of supplementation, blood pressure values did not change in the NAG, while in the AG, we found a decrease of 5.6 mmHg (*p* < 0.05) in mean systolic blood pressure and a decrease of 4.5 mmHg (*p* < 0.05) in diastolic values. There was also a 0.62% increase in muscle mass in the AG vs. the NAG (*p* < 0.05), while the body fat decreased by 1% (*p* < 0.05 in AG vs. NAG). Overall, the AG gained twice the amount of muscle mass and lost twice as much body fat as the NAG. No effects on the mean weighted average heart rate were recorded in the subjects. The results suggest that oral supplementation with arginine can improve blood pressure and body composition, potentially counteracting the stress induced by resistance exercise training. Supplementation with arginine can be a suitable adjuvant for these health benefits in individuals undertaking regular resistance training.

## 1. Introduction

Arginine is classified as a semi-essential or essential conditioned amino acid and plays a central role in the optimal development and maintenance of an individual’s health. However, there are certain situations in which the internal synthesis of arginine cannot be achieved, and its external supplementation is necessary [1,2,3].

The first stage in the arginine synthesis process is citrulline production, from glutamine and glutamate, in the intestinal epithelial cells [4]. Subsequently, citrulline is extracted from the circulation by the proximal renal tubular and converted to arginine, which is then returned to the circulation [5]. Thus, small intestine pathologies (e.g., small intestine resection) or renal function impairment can lead to a decrease in endogenous arginine synthesis and an increase in external demand [6,7].

The intestinal–renal axis is the primary mechanism of internal arginine synthesis, but smaller amounts are also produced in many other cells at a lower level. Arginine production ability increases under circumstances that induce the activity of inducible nitric oxide synthase (iNOS) [8]. Citrulline is a by-product of the catalytic reaction of nitric oxide synthase (NOS) and enters the citrulline–nitric oxide pathway (NO) or arginine–citrulline pathway, leading to arginine formation [9]. Therefore, in many cells, citrulline can replace arginine under certain circumstances in the NO synthesis process [10]. However, the substitution is not optimal in large quantities as citrulline accumulates alongside nitrates and nitrites in NO-producing cells [11].

A phenomenon that is still yet to be explained is the L-arginine paradox, which means that, despite the fact that nitric oxide synthases are saturated with exogenous L-arginine, the amino acid creates no mediated biological effects through the activation of guanylyl cyclase. Among the hypotheses for this phenomenon, we have the vasodilatory role of insulin, the secretion of which is induced by L-arginine, or the fact that NOS activity is determined by neither the extracellular nor intracellular concentration, but rather by the amount of amino acids carried across the plasma membrane. Nakaki T. and Hishikawa K. consider, in their study, that NOS endogenous inhibitors may play a role in the reduced enzyme sensitivity to L-arginine, or that L-citrulline could be an important NOS inhibitor, which consequently leads to L-arginine–L-citrulline competition [12,13].

Oral arginine supplementation in patients with mild arterial hypertension leads to increased citrulline plasma levels and total antioxidant status [14,15]. Elevated plasma levels of l-arginine stimulate NO biosynthesis and decrease oxidative stress [16]. Arterial hypertension accelerates endothelial damage by promoting atherosclerosis, especially in the coronary arteries [17]. Considering that NO is produced from l-arginine, hypotheses have been put forward that the hypertension etiology may be partly due to l-arginine insufficiency [18].

A study that included hypertensive patients treated with oral arginine supplementation (6 g per day in three doses for four weeks) showed a marked improvement in angina pectoris class (graded according to the Canadian Cardiovascular Society), mean blood pressure values and quality of life. Improved values were also recorded in maximal forearm blood flow (FBF), plasma l-arginine levels and cyclic guanosine monophosphate (cGMP) [19]. The beneficial impact of arginine supplementation on blood pressure levels is also supported by a Polish study which found a decrease in blood pressure values in patients treated with oral arginine supplements at 12 g per day [19]. It is essential to mention whether lifestyle changes have been implemented in addition to arginine supplementation. Although often neglected, lifestyle correction improves cardiovascular risk factors and therapeutic adjuvants [20,21]. Recently, research has focused on non-medical therapies for acute pathologies such as stroke. For example, in stroke, non-pharmaceutical therapies (therapeutic hypothermia, certain medical gases, acupuncture) are considered promising strategies for the future [22]. Perhaps the current wave of research will demonstrate the benefits and maximum potency of non-pharmacological therapies, reducing the need for pharmacological therapies in the future.

The endothelial lining of the penis’ corpus cavernosum is significantly vulnerable to free radical action and atherosclerotic plaque formation. As is well-established, this area is more sensitive to these hazards than any other in the body [23]. A poor diet can accelerate the normal age-related progression of endothelial impairment. Age predicts endothelial impairment, so flow-mediated dilation (FMD)—measured endothelial function—decreases with advancing age in healthy individuals [24,25]. In contrast, muscular strength is inversely correlated with diabetes mellitus, arterial hypertension, cardiovascular disease and mortality [26]. These associations are independent parameters of the subjects’ cardio-respiratory fitness [27].

Paradoxically, strength training seems to increase arterial stiffness, a predictor for developing arterial hypertension and an independent risk parameter for cardiovascular morbi-mortality [28,29]. Currently, these data are hypotheses because the accuracy of the relationship between muscular strength and arterial stiffness remains undefined [30].

Considering the hypothesis that resistance exercise leads to arterial stiffness to be true, we evaluated the impact of arginine supplementation on vascular elasticity in healthy individuals who practice regular resistance exercise training.

## 2. Materials and Methods

### 2.1. Profile of Study Participants

Seventy-four non-smoking males between 30 and 45 years old, who had been performing regular resistance exercise training at least three times a week for more than five years, were enrolled in several fitness facilities in Bucharest for a period of two years (1 June 2020–1 June 2022). The initial goal was to create mixed study groups regarding gender, but a lack of effective enrollment of female participants led to our inability to form a representative group. Thus, only male patients were evaluated. The inclusion criteria were: age (between 30 and 45 years old), no personal or familial history of arterial hypertension, type 2 diabetes mellitus, cardiovascular or renal disease and the ability to fully understand the informed consent. Subjects under 18, those without the ability to understand informed consent and subjects diagnosed with cardiovascular disease during initial assessments were excluded. Four of the enrolled participants were excluded from the study after the initial medical assessments, so the final number of participants was 70. There were no dropouts during the study. In our opinion, this was due to certain inclusion criteria (exercise track record, which ensured a disciplined type of participant), the pre-enrolment counseling sessions (describing the study in detail and the importance of continuous participation) and last but not least, the minimal impact the study had on the participants’ daily schedule.

### 2.2. Ethical Considerations

After receiving all the appropriate information and training, the subject’s participation in the study was voluntary. Patient data protection was ensured through pseudo-anonymization. Each participant was assigned an identifier, which had the role of separating personal information from the study’s data collection. No patients were at risk of physical, social, psychological or legal harm during the study. The study was conducted in accordance with the Declaration of Helsinki and approved by the Ethics Committee of Carol Davila University of Medicine and Pharmacy, Bucharest, Romania for studies involving humans.

### 2.3. Initial Assessment

A clinical evaluation was performed at baseline consisting of blood samples (blood count, creatinine, liver enzyme levels, ionogram, lipid profile, blood sugar, uric acid, erythrocyte sedimentation rate, fibrinogen, c reactive protein, urine test), 24 h blood pressure monitoring, electrocardiogram recording and transthoracic echocardiography.

Body composition was assessed using a 4-electrode (feet, hands) weight scale with bioelectrical impedance analysis. In addition, pulse wave analysis was completed using a device based on the oscillometric principle.

### 2.4. Randomisation Process and Six-Month Evaluation

After the baseline evaluation, patients were randomly split into two groups: one group consisted of 35 males selected to take an oral supplement of 5 g of arginine powder before each resistance training session for six months (arginine group, or AG). The other group of 35 subjects was a control group that was assigned to take 5 g of placebo powder (non-arginine group, or NAG). They were not allowed to consume any other oral or injectable performance-enhancement supplements during this period. The dose of arginine administered to the study’s participants was decided considering the fact that pre-workout arginine supplements have a standard measure of 5 g and that current studies demonstrate benefits in cardiovascular disease patients at doses between 6 and 30 g [31,32].

After six months, a clinical and paraclinical assessment consisted of clinical evaluation, blood samples, resting ECG, 24 h blood pressure monitoring, body composition assessment and pulse wave analysis.

### 2.5. Statistical Data Analysis

Statistical analysis was performed via the F-test, *t*-test and chi-squared. Evaluation of the correlations of the variables was performed by scatter graphs. Results with *p* < 0.05 were considered statistically significant. All statistical analysis was performed using the SPSS program and Microsoft Excel.

## 3. Results

At baseline (Table 1), the mean age was 38.77 ± 4.1 years for the AG and 34.42 ± 4.9 years for the NAG. The initial mean weight for the AG was 77.97 ± 4.66 kg compared to the NAG mean weight of 82.02 ± 9.34 kg. The mean muscle mass percentage was high in both groups—38.42% in the NAG, and 40.45% in the AG—with a mean total body fat of 22.46% in the NAG and 21.28% in the AG.

In the NAG, we observed improvements with statistical significance in muscle mass and heart rate compared with the initial values (*p* < 0.001) (Table 2 and Table 3).

In the AG study group, the 6-month evaluation revealed statistically significant (*p* < 0.001) improvements in muscle mass, total fat, systolic and diastolic blood pressure, heart rate and pulse wave velocity. Consequently, after taking a dose of 5 g arginine, the participants had an increase in muscle mass and a decrease in total body fat, systolic and diastolic blood pressure, heart rate and pulse wave velocity (Table 4 and Table 5).

At the six-month reevaluation, both groups had a slight increase in body weight (mean 82.37 Kg in the NAG and 78.05 Kg in the AG) with increases in muscle mass and a decrease in fat percentage in both groups (Table 2 and Table 4).

Using a paired sample *t*-test (Table 3 and Table 5), a comparison was made between the initial assessment and the six-month assessment. Both groups showed a slight variation in body weight with no statistical significance (*p* > 0.05). In the NAG, there was a 0.5% reduction in the total fat (*p* < 0.05) compared to the AG, which had a 1% reduction in the total body fat (*p* < 0.05). In addition, in the NAG, there was a 0.3% increase in the total muscle mass (*p* < 0.05) compared to the 0.62% increase in the AG (*p* < 0.05).

In the NAG, following the 24 h blood pressure monitoring, systolic average blood pressure values decreased with a mean of 0.22 mmHg (*p* = 0.694), diastolic average blood pressure decreased by 0.05 mmHg (*p* = 0.896) and there was a mean pulse average reduction of 1.5 beats per minute (*p* < 0.05). Meanwhile, in the AG, the 24 h blood pressure monitoring systolic average blood pressure values decreased with a mean of 5.6 mmHg (*p* < 0.05), diastolic average blood pressure decreased by 4.5 mmHg (*p* < 0.05) and there was a mean pulse weighted average reduction of 1.5 beats per minute (*p* < 0.05). In addition, in the NAG, pulse wave velocities decreased by 0.03 m/s compared to the AG, where the reduction was 0.15 m/s (*p* < 0.05).

## 4. Discussion

This study investigated the effects of oral arginine supplementation in healthy individuals performing regular physical activity, primarily resistance training. The results obtained from the research suggest that oral supplementation with arginine improves subjects’ blood pressure profiles. In addition, both groups of patients experienced increases in muscle mass and reductions in the total fat percentage when participating in resistance training; nevertheless, the arginine-supplemented group enhanced their muscle mass gains by twice those of the placebo group.

In terms of body fat, the group supplemented with arginine lost twice the amount of body fat compared to the NAG. There was no recorded effect on the heart rate; both groups experienced the same reduction in the weighted average 24 h pulse rate.

Moss et al. evaluated the effect of l-arginine on blood pressure values in their study. They observed the l-arginine blood transport and the plasmatic activity of amino acids in hypertensive patients and on animal models. In those scenarios, they showed that the erythrocyte cell influx of l-arginine is mediated by the “y+” and “y + L” amino acid transport systems. Meanwhile, the platelet cell influx is mediated only by the “y + L” system. Higher plasma l-arginine levels were found in hypertensive patients compared to the control group. Their study results highlighted for the first time the link between hypertension and “y + L” l-arginine transport inhibition. Reduced transport capacity translates into a lower availability of l-arginine and decreased blood cell NO synthesis [7].

Schlaich et al. analyzed arginine’s capture ability in mononuclear blood cells in a study group of hypertensive patients. The results were then compared with two control groups of healthy volunteers, with and without a hereditary history of arterial hypertension. Intra-arterial supplementation of arginine was performed in the study group with subsequent determination of the forearm blood flow response to acetylcholine and sodium nitroprusside. It was discovered that normotensive patients are at high risk of developing hypertension when deficient in l-arginine transport. Consequently, a defective l-arginine/NO pathway can cause the onset of arterial hypertension. In addition, a recent meta-analysis found clear evidence for oral l-arginine supplements significantly decreasing systolic and diastolic blood pressure [33].

A prospective, randomized, double-blind trial on hypertensive patients assessed endothelium-dependent flow-mediated dilation (FMD) of the brachial artery before and after administering l-arginine. The results were then compared to a placebo group. Although no change was observed in either group in heart rate, blood pressure or baseline diameter, the l-arginine administration improved FMD. Thus, it was concluded that l-arginine supplementation improves forearm blood flow in hypertensive patients [31].

Another study evaluated the role of arginine supplementation on arterial resistance, inflammatory processes and metabolic parameters in patients with multiple cardiovascular risk factors. Patients were assessed for lipid and glycemic profiles, systemic inflammation and renin and aldosterone levels. Their arterial elasticity was evaluated through pulse wave contour analysis. At baseline, no differences were found between the groups on the large artery elasticity index (LAEI). However, the LAEI showed a significant improvement in patients treated with l-arginine supplements. At the six-month evaluation, systemic vascular resistance was lower in this group compared to the placebo one. Meanwhile, the small artery elasticity index (SAEI) did not change in value from baseline. Plasmatic aldosterone levels decreased significantly. This study demonstrated the benefits of arginine supplementation for large artery elasticity in patients with cardiovascular disease. Furthermore, improved arterial elasticity has been associated with lower blood pressure values, decreased vascular resistance and reduced aldosterone levels [34].

Puga et al. evaluated the effects of arginine supplementation on postmenopausal women who perform aerobic exercise and found a significant decrease in diastolic blood pressure values, while no beneficial changes were noted in cardio-inflammatory markers [35]. However, Andrade et al. reported a non-significant effect in terms of muscle improvement of I-arginine supplementations after high-resistance exercise [36].

Morais et al. observed the effects of I-arginine on rats after a single session of resistance training. They found immunomodulatory effects of I-arginine in resolving the resistance to exercise-induced muscle inflammation by reducing the expression of atrogin-1 and MuRF-1 atrophic genes [37]. A meta-analysis found that 4–7 and 8-week chronic arginine supplementation improves aerobic and anaerobic exercise performance, respectively. In addition, arginine improved the fatigue threshold and fatigue-related metabolites, enhancing blood flow and improving muscle power and endurance [38].

Muscle hypertrophy and muscular strength are well-established benefits of habitual resistance exercise (RE) [39,40,41,42]. In addition, sustained resistance training improves fasting glucose, insulin levels, plasma triglycerides and body fat percentages while accelerating the basal metabolic rate [43]. Therefore, healthcare providers recommend including resistance exercises in patients’ daily routines for primary and secondary cardiovascular prevention [43].

Arginine might be used in the future as a possible adjuvant supplement in the primary prevention measures to reduce cardiovascular risk and early-stage hypertension in sedentary patients; however, it is advisable first to initiate lifestyle interventions.

## 5. Conclusions

Our study suggests that oral arginine supplementation improves the body composition and arterial elasticity in healthy individuals performing resistance training exercises. Our results are mainly of interest to high-intensity resistance training athletes seeking to combat the effects of this type of exercise on blood pressure and vascular stiffness. Further studies are needed to evaluate the benefits of these supplements in patients with known arterial hypertension.

## 6. Study Limitations

Limitations of the study include how the groups’ workout programs could not be standardized and the groups were not mixed in terms of gender. All subjects performed regular resistance training exercises at different intensities and weight regimens. Subjects reported no other nutritional supplementation during the research, but their personal dietary intake could not be monitored.

## Figures and Tables

**Table 1 healthcare-11-00182-t001:** Descriptive characteristics of patients enrolled in the study.

	All (Mean ± SD, Min.–Max.) *n* = 70	NAG (Mean ± SD, Min.–Max.) *n* = 35	AG (Mean ± SD, Min.–Max.) *n* = 35
Age (years)	36.6 ± 5.00 (26–46)	34.42 ± 4.92 (26–44)	38.77 ± 4.11 (31–46)
Weight (kg)	80.00 ± 7.61 (67–99)	82.02 ± 9.34 (67–99)	77.97 ± 4.66 (71–91)
Height (cm)	178.44 ± 6.75 (163–193)	177.28 ± 6.39 (163–185)	179.60 ± 6.43 (170–193)
TotalFat (%)	21.87 ± 4.08 (12.8–32.3)	22.46 ± 4.65 (12.8–32.3)	21.28 ± 3.39 (13–26)
Muscle mass (%)	39.44 ± 2.99 (32,4–45,6)	38.42 ± 3.29 (32.4–44.6)	40.45 ± 2.28 (36.4–44.6)
BMI	24.92 ± 2.36 (19–33)	26.05 ± 2.31 (22–33)	23.80 ± 1.82 (19–28)
SysBP (mmHg)	13.92 ± 11.81 (110–156)	133.31 ± 11.82 (111–156)	128.54 ± 11.48 (110–152)
DiastBP (mmHg)	74.42 ± 5.93 (63–84)	75.57 ± 6.27 (63–84)	73.28 ± 5.43 (67–83)
HR (b/min)	65.88 ± 6.21 (50–77)	67.00 ± 6.20 (50–77)	64.77 ± 6.11 (50–71)
PWV (m/s)	6.60 ± 0.40 (5.8–7.3)	6.69 ± 0.30 (5.9–7.3)	6.50 ± 0.46 (5.8–7.3)

NAG = non-arginine group; AG = arginine group; BMI = body mass index; SysBP = systolic blood pressure; DiastBP = diastolic blood pressure; HR = heart rate; PWV = pulse wave velocity.

**Table 2 healthcare-11-00182-t002:** Descriptive characteristics of patients in the NAG at baseline and six months.

Descriptive					
	N	Mean	SD	SE	Coefficient of Variation
Weight _baseline_	35	82.029	9.348	1.580	0.114
Weight _6 months_	35	82.371	7.897	1.335	0.096
Muscle mass _baseline_	35	38.429	3.295	0.557	0.086
Muscle mass _6 months_	35	38.734	3.241	0.548	0.084
TotalFat _baseline_	35	22.466	4.655	0.787	0.207
TotalFat _6 months_	35	21.960	4.352	0.736	0.198
SysBP _baseline_	35	133.314	11.822	1.998	0.089
SysBP _6 months_	35	133.086	11.289	1.908	0.085
DiastBP _baseline_	35	75.571	6.270	1.060	0.083
DiastBP _6 months_	35	75.514	5.442	0.920	0.072
HR _baseline_	35	67.000	6.202	1.048	0.093
HR _6 months_	35	65.486	5.501	0.930	0.084
PWV _baseline_	35	6.697	0.303	0.051	0.045
PWV _6 months_	35	6.660	0.319	0.054	0.048

Sys BP = systolic blood pressure; DiastBP = diastolic blood pressure; HR = heart rate; PWV = pulse wave velocity.

**Table 3 healthcare-11-00182-t003:** Comparative analysis of the placebo group at baseline and six months according to the paired samples *t*-test.

Measure Baseline	Measure 6 Months	t	df	*p*
Weight	Weight	−0.564	34	0.577
Muscle mass	Muscle mass	−8.917	34	<0.001
TotalFat	TotalFat	3.246	34	0.003
SysBP	SysBP	0.397	34	0.694
DiastBP	DiastBP	0.132	34	0.896
HR	HR	4.920	34	<0.001
PWV	PWV	2.130	34	0.040

Note. Student’s *t*-test. SysBP = systolic blood pressure; DiastBP = diastolic blood pressure; HR = heart rate; PWV = pulse wave velocity.

**Table 4 healthcare-11-00182-t004:** Descriptive characteristics of patients in the arginine group at baseline and six months.

Descriptive					
	N	Mean	SD	SE	Coefficient of Variation
Weight _baseline_	35	77.971	4.662	0.788	0.060
Weight _6 months_	35	78.057	4.291	0.725	0.055
Muscle mass _baseline_	35	40.454	2.286	0.386	0.057
Muscle mass _6 months_	35	41.080	2.195	0.371	0.053
TotalFat _baseline_	35	21.286	3.392	0.573	0.159
TotalFat _6 months_	35	20.206	3.344	0.565	0.165
SysBP _baseline_	35	128.543	11.480	1.940	0.089
SysBP _6 months_	35	122.914	9.559	1.616	0.078
DiastBP _baseline_	35	73.286	5.437	0.919	0.074
DiastBP _6 months_	35	68.743	4.955	0.838	0.072
HR _baseline_	35	64.771	6.112	1.033	0.094
HR _6 months_	35	63.257	5.420	0.916	0.086
PWV _baseline_	35	6.509	0.465	0.079	0.072
PWV _6 months_	35	6.351	0.425	0.072	0.067

**Table 5 healthcare-11-00182-t005:** Comparative analysis of the arginine group at baseline and six months.

Paired Samples *t*-Test				
Measure Baseline	Measure 6 Months	t	df	*p*
Weight	Weight	−0.119	34	0.906
Muscle mass	Muscle mass	−12.448	34	<0.001
TotalFat	TotalFat	6.707	34	<0.001
SysBP	SysBP	5.078	34	<0.001
DiastBP	DiastBP	14.034	34	<0.001
HR	HR	5.484	34	<0.001
PWV	PWV	6.644	34	<0.001

Note. Student’s *t*-test. SysBP = systolic blood pressure; DiastBP = diastolic blood pressure; HR = heart rate; PWV = pulse wave velocity.

## Data Availability

The data presented in this study are available on request from the corresponding author. The data are not publicly available due to ethical restrictions.

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
