# Peer review of "Oral Arginine Supplementation in Healthy Individuals Performing Regular Resistance Training"

_healthcare, 2023, doi:10.3390/healthcare11020182_

Round 1

Reviewer 1 Report

Comments to the Authors of manuscript number: healthcare-2120759 entitled “Oral arginine supplementation in healthy individuals performing regular resistance training”.

It is very good study and very interesting which should be published, but after small correction.

1. L 26 what is biology?

2. L38- a small letter

3.L 52, 75, 80, 87 - punctuation mark. Check the whole text

4.L 52 – what is a small intestine pathology?

5. L 66 – what are biological effects?

6. L 67- the reference should be added

7. L 69 – of what this concentration?

8. L 70-71 – some authors and one reference

9. L 83-  what is angina class?

10. L 83 – mean or other type of blood pressure?

11. L 84 – small letters

12. L 91 – what are “ many acute pathologies”? and add the ref

13. L 1000 – what is FMD?

14. L 109 – there should be presented what do Authors want to do according to the hypothesis.

15. L 24 speaks that there were 70 men, but L 113 – 74?

16. L 134- how these functions were determined?

17. L 135 – what inflammation markers?

18. L 142- how this amount of arginine was chosen? Recommendation? Add the ref.

19. L 148 – references should be added

Author Response

Dear Reviewer,

Firstly, I would like to thank you for your time  offered to review our manuscript.

Next, I will respond punctually to the mentions you have adressed us:

  1. We have replaced biology with blood samples.
  2. We have corrected the uppercase character.
  3. We have checked the whole text for punctuation errors.
  4. We have added in paranthesis (intestinal resection).
  5. We have completed the description of the NO biological effects.
  6. The reference was added.
  7. Here, what we meant was that regardless of the concentration value, it plays no role in the NOS activity.
  8. We have corrected the error.
  9. We have corrected the error through a more detailed description.
  10. We have corrected it to mean blood pressure.
  11. The small letter have been corrected.
  12. We have replaced many acute pathologies with stroke, due to the fact that the cited studiy refers to stroke.
  13. The description of FMD (flow meditated dilation) has been added.
  14. We have added a more extensive description of our purposes according to the hypothesis raised.
  15. Four of the enrolled patients were excluded from the study after the initial medical assessment, so the final number of participants was 70. We have completed the description with all these details (L 124-126).
  16. Kidney and liver functions have been replaced with creatine and liver enzyme levels (L 137-138)
  17. Inflammation markers have been replaced with erythrocyte sedimentation rate, fibrinogen and c reactive protein.
  18. The dose of arginine administered to the study's participants was decided considering the fact that pre-workout arginine supplements have a standard measure of 5g and that current studies demonstrate benefits in cardiovascular disease patients at doses between 6-30g - references [31][32].
  19. The reference has been added.

I hope that I was able to answer your questions and that my answers were up to expectations. 

Kind regards,

Maria Pană

Reviewer 2 Report

The study is well written, and analysis is simple and presented in a discreet manner. 

The issues are as follows:

Age group of the study participants has been mentioned different at different places in the manuscript. In Abstract it is 30-45 while in Methodology it is 35-45 and 30-50 in the same paragraph.

It is also mentioned that it took 2 years to recruit the study participants, so what actual was the study period, i.e., start and end dates of the intervention?
It is mentioned that there was a lack of femaleparticipants..... What exactly this means, elaborate.

It appears there was no drop-out in this study, what strategies were done to make sure the retention of participants in the study.
Also, there should be a mention of the study area elaborating where were the recruitments done from, i.e., from gymnasiums, from spas or any other places?

Can there be a recall bias in this study? If so, mention in the limitations

Author Response

Dear Reviewer, 

Thank you for your time offered to review our manuscript.

I will respond punctually to the issues you have identified:

Regarding the age of the participants, a write error occured. We have corrected the age of the participants (30-45). The inclusion criteria was 30-50 years old.

The start and end dates of the intervention are 01.06.2020-01.06-2022.

Regarding the lack of female participations, we encountered very few women who met the inclusion criteria and none of them wanted to participate in the study (most of them did not provide a concrete reason, others because they did not want to take any oral supplements.)

Fortunately, we did not have any dropouts from the study participants. First, the minimum 5-year sport track record we introduced as an inclusion criterion ensured a disciplined type of participant. Before enrollment, we conducted counselling sessions with them in which we explained the purpose of the study in great detail and insisted on the importance of participation until the end. Finally, the intervention on their daily routine was minimal, only the medical evaluations and the arginine/placebo supplementation, but without a major change of their usual schedule. 

The study was elaborated in gyms. 

In our opinion, there isn't any recall bias, considering that the study is based on objective measurements of the parameters.

I hope that I was able to answer all your questions and that my answers were up to expectations.

Kind regards,

Maria Pană

Round 2

Reviewer 2 Report

Lines 25 and 120 still contains age criteria different from the one mentioned in responses to the reviewer's suggestions.
The reason for no dropouts should be mentioned in the manuscript text.
The manuscript otherwise looks fine to be accepted

Author Response

Dear Reviewer,

We have corrected the age criteria accordingly. 

We have added the reasons for no dropouts during the study (L 127-131).

Thank you for your time and valuable suggestions.

Sincerely, 

M.P.